# VEGFC Gene Expression Is Associated with Tumor Progression and Disease-Free Survival in Cutaneous Squamous Cell Carcinoma

**DOI:** 10.3390/ijms25010379

**Published:** 2023-12-27

**Authors:** Omar García-Pérez, Leticia Melgar-Vilaplana, Inés Sifaoui, Aleksandra Śmietańska, Elizabeth Córdoba-Lanús, Ricardo Fernández-de-Misa

**Affiliations:** 1Research Unit, Hospital Universitario Nuestra Señora de Candelaria, 38010 Santa Cruz de Tenerife, Spain; omargp6@gmail.com; 2Instituto Universitario de Enfermedades Tropicales y Salud Pública de Canarias (IUETSPC), Universidad de La Laguna, 38200 San Cristóbal de La Laguna, Spain; isifaoui@ull.edu.es; 3Pathology Department, Hospital Universitario Nuestra Señora de Candelaria, Ctra. Gral. del Rosario, 145, 38010 Santa Cruz de Tenerife, Spain; leticiamvfreedom@gmail.com; 4Faculty of Pharmacy, Medical University Wroclaw, 50-556 Wroclaw, Poland; aleksandra.smietanska98980@gmail.com; 5Centro de Investigación Biomédica en Red, CIBERINFEC, Instituto de Salud Carlos III, 28029 Madrid, Spain; 6Dermatology Department, Hospital Universitario Nuestra Señora de Candelaria, Ctra. Gral. del Rosario, 145, 38010 Santa Cruz de Tenerife, Spain

**Keywords:** cutaneous squamous cell carcinoma, VEGFC, angiogenesis, lymphangiogenesis, biomarkers

## Abstract

Cutaneous squamous cell carcinoma (CSCC) is one of the most common cancers in the skin. CSCC belongs to the non-melanoma skin cancers, and its incidence is increasing every year around the world. The principal routes of tumor progression are related to angiogenesis and lymphangiogenesis. In this study, we assess the gene expression of the relevant biomarkers of both routes in 49 formalin-fixed paraffin-embedded (FFPE) CSCC samples in an attempt to determine a molecular profile that correlates with the tumor progression and disease-free survival (DFS). The results were enhanced by a posttranscriptional analysis using an immunofluorescence assay. Overexpression of the vascular endothelial growth factor C (*VEGFC*) gene was found in patients with tumor progression (*p* = 0.022) and in patients with perineural invasion (*p* = 0.030). An increased expression of protein VEGFC in samples with tumor progression supported these results (*p* = 0.050). In addition, DFS curves showed differences (*p* = 0.027) for tumors with absent-low *VEGFC* expression versus those with high levels of *VEGFC* expression. No significant influence on DFS was detected for the remaining analyzed genes. *VEGFC* expression was found to be a risk factor in the disease progression (HR = 2.675; 95% CI: 1.089–6.570; *p* = 0.032). Our main results suggest that *VEGFC* gene expression is closely related to tumor progression, DFS, and the presence of perineural invasion.

## 1. Introduction

The incidence of skin cancer is increasing worldwide, mainly due to chronic exposure to sunlight, climatic changes, and individual and social conditions [1]. Skin cancer includes two groups: malignant melanoma (MM) and non-melanoma skin cancer (NMSC). NMSC is the most common cancer around the world [2]. In fact, in the UK, roughly 156,000 new cases are diagnosed every year. It means nearly 430 new cases every day (2016–2018). Basal cell carcinoma (BCC) accounts for 80% of NMSC, whereas cutaneous squamous cell carcinoma (CSCC) accounts for the remaining 20% [3].

NMSC usually exhibits favorable clinical behavior, showing only local involvement. But occasionally, it spreads to regional nodes or even develops distant metastases. Ten-year survival after surgery exceeds 90% for CSCC but drops severely when metastases occur [4]. The frequency of lymph node metastases is around 4%, and the mortality rate reaches nearly 2%. But given its high frequency, invasive CSCC has a significant impact on morbidity and overall mortality [5]. Well-established prognostic factors for metastatic disease inferred from the primary tumor are tumor diameter, location, the degree of histological differentiation and histologic features, the depth of the tumor (thickness or level of invasion), the existence of perineural invasion, and the presence of lymphatic or vascular involvement [6,7,8]. Tumors at higher risk of recurrence, metastases, or specific death are those larger than 4 cm at any location that show a poor degree of differentiation and a desmoplastic pattern, are deeper than 6 mm or invade beyond the subcutaneous fat, and show significant perineural involvement and lymphatic/vascular involvement [8]. Currently, CSCC is the second most common cause of death from skin cancer after melanoma and causes most deaths from skin cancer in people older than 85 years [9]. In fact, there are areas in the US where the mortality rate due to CSCC compares to that of renal cancer, oropharyngeal cancer, and melanoma [10]. 

In tumorigenesis, cell growth requires the development, differentiation, and growth of new blood and lymphatic vessels for tumor development and spread [3]. These events are known as angiogenesis and lymphangiogenesis. Both are crucial for tumor cell dissemination [11], playing a fundamental role in angiolymphatic invasion which has been correlated with poor prognosis and survival in other skin cancers such as MM [12]. In the development of CSCC, the angiogenic imbalance occurs early, and angiogenesis has been reported to be linked to tumor progression [13]. In reference to lymphangiogenesis, Moussai et al. (2011) found by qPCR and immunofluorescence that there was a higher density of lymphatic endothelial vessels in the dermis immediately adjacent to CSCC nests, due to the expression of VEGFC [14].

The main angiogenic and lymphangiogenic inductors are the vascular endothelial growth factor (VEGF) and its receptors (VEGFR2, VEGFR3). Their main functions are involved in growth, migration, and vascular-lymphatic permeability [15,16]. It was previously detected by immunohistochemistry that VEGF was expressed in the endothelial cells of the blood vessels of both the adjacent skin and the CSCC tumor [17]. In fact, the increased gene expression of *VEGFC* in skin closely adjacent to primary CSCC was confirmed by qPCR [14]. In another skin cancer, MM, the expression of VEGFC, VEGFR2, and VEGFR3 was reported to be significantly higher in the metastatic tissues [18]. Furthermore, in MM patients, a higher *VEGFR3* expression was found in the positive sentinel node when compared to negative ones, suggesting that *VEGFR-3* may play a role in the progression of MM [19]. In other tumors, the existence of high serum levels of VEGF has been related to poor prognosis [20,21,22,23]. Moreover, in oral squamous cell carcinoma, the positive gene expression of *VEGFA* could be used as a prognostic risk biomarker [24]. 

Another protein involved in lymphangiogenesis is lymphatic vessel endothelial hyaluronan receptor 1 (LYVE-1). This lymph-specific hyaluronan receptor has been characterized as an important marker for the lymphatic vessels [25]. Some studies showed that LYVE1 immunostaining can be detected in MM cells within lymphatics but is not reliable in predicting MM metastasis, as it does not detect metastatic spread in more than two-thirds of patients with regional node metastases [26]. Instead, high levels of expression of LYVE1 and podoplanin in the primary tumor were correlated with lymph node metastasis in oral squamous cell carcinoma [27]. 

The transcription factor Prospero homeobox 1 (PROX1) [28] is decisive for the embryonic development of the lymphatic system, liver, retina, pancreas, etc. [29]. It has been seen that PROX1 is related to hypoxia-inducible factor 1-α (HIF1α), which is a regulatory factor of hypoxia. Furthermore, a high expression of PROX1 in the primary tumor of esophageal squamous cell carcinoma was reported to contribute to shorter survival and to be related to local lymph node and distant metastasis [30].

The study of the expression of these angiogenic and lymphangiogenic promoters in primary tumor samples may increase the possibilities of an early detection of metastasis in patients with CSCC. 

The current study focuses on patients with CSCC with the aim to (1) analyze the relationship between the main features of CSCC and the progression of the disease; (2) assess the gene expression levels of the relevant biomarkers of angiogenesis and lymphangiogenesis in the primary tumor; and (3) explore the prognostic information that the gene expression data provide in relation to tumor progression in patients with CSCC.

## 2. Results

### 2.1. Clinicopathological Features of CSCC and Disease Progression

Overall patients showed a median age at diagnosis of 75 years, but those showing tumor progression were older (*p* = 0.007). Tumors with high Breslow (thickness) (7.0 mm, 6.0–11.0 vs. 3.0 mm, 2.0–5.0; *p* = 0.006) and diameter indexes (2.7 mm, 1.5–12.0 vs. 1.10 mm, 0.80–2.20; *p* = 0.007) were more likely to develop disease progression. Tumors showing Clark index IV–V exhibited a higher risk of developing disease progression than CSCC with Clark I–III (*p* = 0.004). In the same way, tumors exhibiting perineural invasion progressed more frequently compared to those tumors without perineural compromise (*p* = 0.010). The main clinicopathological characteristics of these patients are shown in Table 1.

Table 2 shows the effect of the tumor progression on various clinicopathological variables. Tumor progression was associated with age (≤75 vs. >75 years) (OR = 3.850, 95% CI: 1.078–13.751; *p* = 0.038), thickness (≤6 vs. >6 mm) (OR = 9.333, 95% CI: 1.447–60.213; *p* = 0.019), Clark level (I–III vs. IV–V) (OR = 2.224, 95% CI: 1.082–4.571; *p* = 0.030), and perineural invasion (absent vs. present) (OR = 9.300, 95% CI: 1.613–53.618; *p* = 0.013).

### 2.2. Angiogenic and Lymphangiogenic Gene Expression Profile in Patients with CSCC

The results of the relative gene expression of *VEGFA*, *VEGFR2*, *VEGFC*, *VEGFR3*, *LYVE1*, and *PROX1* attending to the main clinicopathological variables of the patients are shown in Table 3. The presence of an increased *VEGFC* gene expression in CSCC tumor samples with perineural invasion is remarkable compared to those without perineural invasion (0.89 ± 0.29 vs. 0.37 ± 0.07; *p* = 0.030). On the other hand, *VEGFR3* gene expression was higher in patients with the lowest median age at diagnosis ≤ 75 years compared to those with the oldest age > 75 years (1.83 ± 0.55 vs. 1.00 ± 0.74; *p* = 0.043).

The gene expression profile showed a relatively higher expression of *VEGFC* in patients with disease progression compared to those without it (*p* = 0.022). (Figure 1). The posttranscriptional protein immunofluorescence detection performed in CSCC FFPE tissues confirmed that the expression of VEGFC was significantly higher in primary tumors corresponding to patients with disease progression compared to those without it (*p* = 0.05) (Figure 2).

### 2.3. Prognostic Information Derived from the Gene Expression Profile in the Primary Tumor

Disease-free survival (DFS) curves showed statistically significant differences (log-rank test *p* = 0.027) for high levels of *VEGFC* expression (75.17 months; 95% CI: 39.97–110.38) versus low and absent *VEGFC* expression (115.44 months; 95% CI: 96.31–134.58) (Figure 3 and Table 4). No significant influence on DFS was detected for the remaining analyzed genes.

Cox regression analysis, considering tumor diameter and perineural invasion as correction factors for each gene, was performed. *VEGFC* gene expression seems to have a risk role (HR = 2.657, 95% CI: 1.089–6.570; *p* = 0.032) in the disease progression (Table 4).

## 3. Discussion

Angiogenesis and lymphangiogenesis are essential for the growth of blood and lymphatic vessels, generating an essential homeostasis for the development of any organism. Therefore, the imbalance of these processes leads to the appearance of multiple diseases [31,32]. Angiogenesis and lymphangiogenesis are also important in tumor proliferation and progression, participating in the development of distant metastasis [33,34]. In the current study, the relationship found between tumor progression and the main characteristics of the disease such as the Breslow index (thickness), Clark level, tumor diameter, and perineural invasion, among others, agree with the results of previous series. Remarkably, the expression of *VEGFC*, a gene involved in angiogenesis/lymphangiogenesis, was found significantly increased in samples of primary CSCC with disease progression.

According to the eighth edition of the AJCC [35], this study further strengthens certain risk factors that are included in the staging system of CSCC. The Breslow index and the tumor diameter are two of them [36]. Our data corroborate the importance of these factors, since the greater the thickness of the primary tumor, the higher the probability of disease progression and, therefore, the worse prognosis. Similarly, Clark levels above III were also related to disease progression. Another relevant histological variable is perineural invasion. Perineural invasion predicts a poor prognosis in several cancers such as pancreatic [37], gastric [38], and head and neck CSCC [39]. In the present study, it was found that patients with tumors presenting perineural invasion had a significantly higher risk of disease progression. Interestingly, patients who developed CSCC progression were older at diagnosis (median > 82 years) than patients who did not show tumor progression. Age has a direct correlation with tumor progression; for example, previous research has revealed that individuals older than 75 years are at an increased risk of developing metastatic CSCC [40]. Finally, we found no differences between disease progression and tumor differentiation, location, or the gender of patients. We cannot discard that this lack of association is probably due to the sample size.

The main objective of this work was to study the correlation between the expression of genes involved in angiogenesis/lymphangiogenesis and the presence of disease progression. We found *VEGFC* expression to be significantly increased in samples of primary CSCC with disease progression. These results support previous findings which suggest that *VEGFC* expression is associated with tumor progression. This relationship may promote the metastatic process in different tumors through angiogenesis and lymphangiogenesis processes. For example, Kodama et al. (2008) studied *VEGFC* expression in different gastric carcinoma cell lines, finding that *VEGFC*-expressing tumor cells played an important role in the progressive growth of gastric carcinoma in humans through autocrine and paracrine mechanisms [41]. In addition, increased protein expression of VEGFC has also been observed in tumors such as ovarian carcinoma [42] and breast cancer [43]. However, studies carried out on patients with CSCC are scarce. Moussai et al., 2011, confirmed by qPCR the presence of high levels of *VEGFC* in the skin adjacent to the tumor [14]. This fact could indicate coordination of the metastatic process in the lymphatic vessels associated with the tumor [14]. It has also been reported that *VEGFC* expression is related to tumor progression in head and neck squamous cell carcinomas [44]. Except for *VEGFC*, we could not establish a relationship between the other genes studied and the progression of the disease. Furthermore, overexpression of the *VEGFC* gene acts as a risk factor in CSCC. In other skin cancers such as melanoma, it has also been seen that the protein expression of VEGFC is an important risk factor and predictor in this cancer [45].

Another less well-known route of tumor dissemination is perineural invasion. In this research, the overexpression of *VEGFC* was significantly detected in CSCC tumors with perineural invasion. Although *VEGFC* is known to be mainly associated with the growth of lymphatic vessels during lymphangiogenesis, it could indirectly influence perineural invasion, since as cancer cells spread through the lymphatic system, they could encounter nerves and infiltrate the perineural space. Furthermore, sympathetic innervation promotes the development of the tumor microenvironment and tumor growth due to its own sympathetic signaling that is capable of inducing an angiogenic change through VEGF levels [46]. As far as we know, this is the first study analyzing *VEGFC* gene expression in CSCC samples. However, other investigators [47] reported this same finding in oral squamous cell carcinoma.

As strengths of our study, we must highlight the presence of a well-characterized cohort of patients with CSCC. The main prognostic parameters [35] were adequately recorded in a large proportion of patients that underwent prolonged follow-up. However, our research also has limitations. Although our results clearly underscore the association between *VEGFC* and the tumor progression of CSCC, the small number of patients with disease progression or perineural invasion are important limitations. In the case of CSCC patients with perineural invasion, Campoli et al. (2014) observed that only 4.6% of 753 patients with CSCC had perineural invasion [48]. In every single retrospective study, a larger series would be advisable as a sample size, but it is very important to consider that it is difficult to address. Nevertheless, an independent cohort with a larger series is needed to confirm these initial findings. The convenience sampling model may have biased the recruited patients. Another limitation to consider is that in the multivariate analysis, only two variables, diameter and perineural invasion, were considered since they are the only two variables dependent on the primary tumor that are included in the TNM classification for patients with CSCC. However, a large sample of patients is needed to evaluate other relevant clinical variables. 

Although there are still questions regarding tumor development in CSCC, the present investigation provides valuable information to the field of study since there is little research on the differential expression of genes involved in angiogenic and lymphangiogenic processes in CSCC. Furthermore, it is important to mention that this research confirms that factors such as age and the thickness and diameter of the tumor, as well as perineural invasion, have highly significant associations with the progression of CSCC.

## 4. Material and Methods

### 4.1. Patients and Study Samples

Using the convenience sampling model, 49 formalin fixed paraffin embedded (FFPE) samples of primary CSCC, corresponding to 49 patients, were included in this study. This series comprises 16 tumors with disease progression and 33 without progression. These patients were diagnosed by the Dermatology Department at Hospital Universitario Nuestra Señora de Candelaria (HUNSC) (Santa Cruz de Tenerife, Spain), and their follow-up was longer than 24 months. The samples were stored at the Pathology Department (HUNSC). The study was approved by the local Ethics Committee (C.P. MO—C.I. PI-57/17 and C.P. MO—C.I. PI-39/14). 

### 4.2. RNA Isolation for Gene Expression Assays

Six of the most relevant genes in the development of angiolymphatic and lymphatic vessels out of a set of genes previously reported in the literature [14,17,18,19,25,26,27,28] were selected for analysis. These genes were *VEGFA*, *VEGFR2*, *VEGFC*, *VEGFR3*, *LYVE1*, and *PROX1* (Table 5).

From each CSCC tissue paraffin block, 3 to 6 sections around 5–10 µm were sliced for histochemical detection in routine procedures. Then, the first slice was discarded to avoid contamination, and all sections were macrodissected before RNA purification. “RNeasy FFPE kit” (Qiagen) was used for the RNA isolation from FFPE tissue, with a first step that included a deparaffiniser solution. RNA concentration was measured using the NanoDrop ND-1000 spectrophotometer (ThermoFisher Scientific Inc., Waltham, MA, USA). Once the RNA was transcribed, the integrity of the RNA was confirmed by the amplification of the human *ACTB* gene by conventional PCR (endogenous control), as previously published [49]. “High-Capacity cDNA Reverse Transcription Kit” was used for the synthesis of the second strand of RNA (ThermoFisher Scientific Inc., Waltham, MA, USA). 

### 4.3. Target Gene Expression by qPCR

The obtained cDNA was preamplified using a “TaqMan PreAmp Master Mix”, which amplifies small amounts of cDNA without introducing amplification bias. The concentration used to carry out the preamplifications was 100 ng/µL and run in a thermocycler under the following conditions: 14 cycles of 95 °C for 15 s and 60 °C for 4 min (after activation of the polymerase at 95 °C for 10 min). Once the reaction was finished, the amplified product obtained was diluted 1:5 in water for molecular biology. TaqMan specific predesigned probes (Thermo Fisher Scientific Inc., Waltham, MA, USA) (Table 5) for all the target genes were used at a concentration of 0.05× in a final 10 µL pre-amp reaction.

Subsequently, “TaqMan Gene Expression Master Mix” (10X) (ThermoFisher Scientific Inc., Waltham, MA, USA) with the correspondent TaqMan specific probes above cited, were used for the relative gene expression quantification reaction in a final volume of 10 µL and set up in a Step One Plus (Thermo Fisher Scientific Inc., Waltham, MA, USA) real-time PCR detection machine. Expression data were calculated using 2(-Delta Delta C(T)) method [50]. The reference genes used to normalize the variations between samples were hypoxanthine phosphoribosyltransferase 1 (*HPRT1*) and transferrin receptor-1 (*TFRC*) (García-P, et al., 2021) [51]. Every sample was run in triplicate, and a non-template control was included in every reaction. A control sample was used as an internal calibrator and run in every plate to normalize for inter-plate variation. 

### 4.4. Immunofluorescence Staining

After histological routine observation of eight FFPE samples (5 tumor samples without disease progression and 5 with progression), three sections of 5–6 µm of the tumor were sliced using the Reichert-Jung microtome for the immunofluorescence assay. After deparaffinization using xylol, washes and rehydration of the slides by immersing them in decreasing concentrations of EtOH up to H_2_O were performed. Later, CSCC slide samples were pretreated with 10 mM citrate buffer (pH 6) at 90 °C for 5 min for antigen retrieval. Blocking of nonspecific binding was performed in 0.5% casein (Sigma-Aldrich, St. Louis, MO, USA) for 1 h at room temperature (RT). Slides were then incubated with 1:200 diluted polyclonal rabbit VEGFC antibody (bs-1586R, Bioss Antibodies, ThermoFisher Inc., Waltham, MA, USA) for the assessment of VEGFC expression. These incubations were performed overnight at +4 °C. After washing up with PBS, slides were incubated with 1:100 diluted goat antirabbit antibody (F0382-1ML; Sigma-Aldrich) for 1 h at RT. Additional washings in PBS were performed. Finally, the slides were marked and mounted with Fluoroshield with DAPI (Sigma-Aldrich).

The images were obtained using an inverted confocal microscope Leica DMI 4000B with LASX Office 1.4.5 software, with 405 nm and 488 nm laser, and a Leica ACS APO 40×/1.15 objective. Also, oil objective was used. The microscope images were analyzed using the ImageJ 1.54g software (NIH, Image Processing and Analysis software, https://imagej.nih.gov/ij/ accessed on 18 October 2023).

### 4.5. Statistical Analysis 

Chi^2^, Fisher exact, and binary logistic regression were performed for categorical variable analysis to assess the existence or not of the disease progression and the relative expression of the studied genes. Continuous variables were compared using Mann–Whitney tests. Relative gene expression was Log2-transformed. Patients’ survival was estimated using the Kaplan–Meier method and compared by means of the log-rank test. For this analysis, the variable gene expression was divided into low or absent expression versus high expression level by the median value observed. Cox proportional hazard regression model was used for multivariant analysis in order to estimate the hazard ratio (HR) and 95% CI. Significance level was set to *p* < 0.05 for all tests. Statistical analyses were performed using SPSS v.26 (IBM Corp, Armonk, NY, USA) and graphics with GraphPad Prism 9 (GraphPad Software, San Diego, CA, USA). 

## 5. Conclusions

The angiogenesis and lymphangiogenesis are crucial processes that facilitate tumor development and expansion. Our main results suggest that the high expression of *VEGFC* is closely related to tumor progression and perineural invasion. Furthermore, the overexpression of *VEGFC* is associated with a decreased DFS. In addition, an older age, increased tumor thickness, Clark level, and the existence of perineural invasion are correlated with the tumor progression. The current results highlight the possibility of identifying biomarkers that increase our knowledge about the disease’s prognosis, thus improving our approach to patients with cutaneous squamous cell carcinoma.

## Figures and Tables

**Figure 1 ijms-25-00379-f001:**
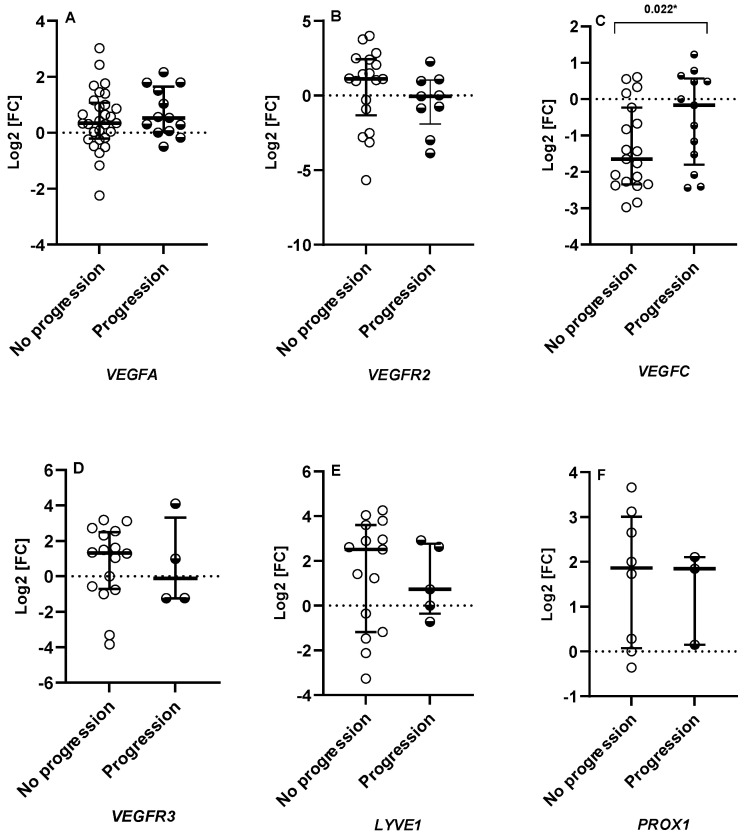
Relative expression of the studied genes in relation to the existence of tumor progression (median and interquartile range). Relative expression data are represented using Log2 Fold-Change (Log2FC). (**A**–**F**) indicate the following: *VEGFA*, *VEGFR2*, *VEGFC*, *VEGFR3*, *LYVE1*, and *PROX1*. * *p*-value < 0.05 was considered significant.

**Figure 2 ijms-25-00379-f002:**
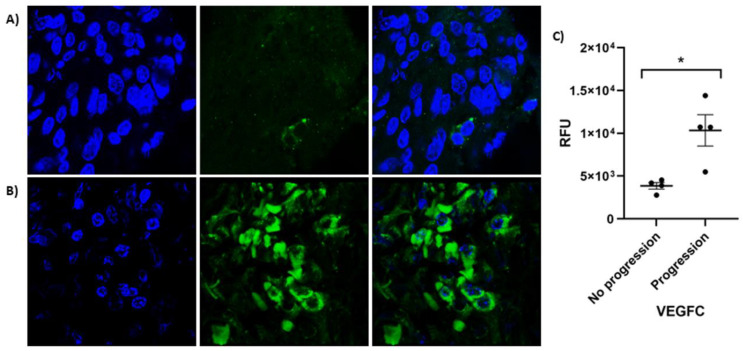
VEGFC protein in CSCC tumors detected by immunofluorescence. In blue, DAPI stain cell nuclei. In green: polyclonal rabbit VEGFC antibody (bs-1586R) that recognizes endogenous levels of total VEGFC protein. Merge is the superposition of images with both stains. (**A**) CSCC tissues without tumor progression. (**B**) CSCC tissues with disease progression. (**C**) Relative fluorescent units (RFU) of VEGFC protein in the tumors in relation to the existence or not of disease progression. * *p*-value < 0.05 was considered significant.

**Figure 3 ijms-25-00379-f003:**
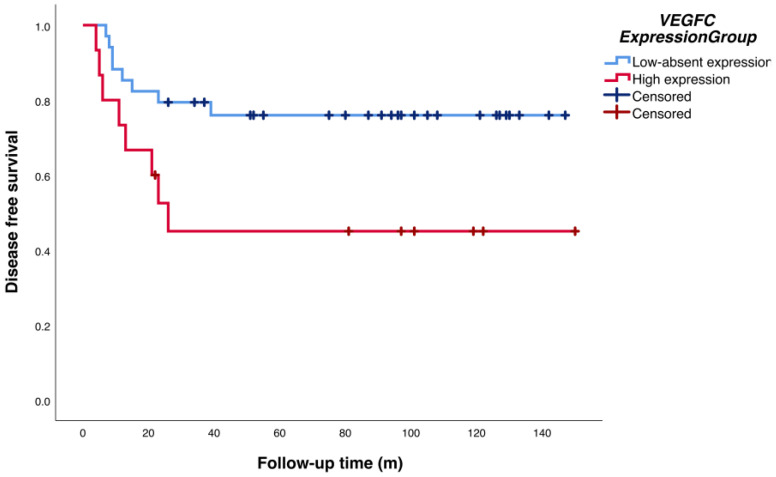
Kaplan–Meier curves plot DFS probability by the high-low/absent gene expression of *VEGFC*. DFS: disease-free survival.

**Table 1 ijms-25-00379-t001:** Clinicopathological characteristics of patients with CSCC based on the existence of disease progression.

	CSCC Cases N = 49	No Disease Progression (*n* = 33)	Disease Progression (*n* = 16)	*p*-Value
Age (y) ^±^	75 (67–81.5)	72 (66–77)	82 (72–87)	**0.007 * (‡)**
**Gender *n* (%)**				
Female	21 (42.9)	14 (42.4)	7 (43.8)	1.000 (ϕ)
Male	28 (57.1)	19 (57.6)	9 (56.2)	
**Localization *n* (%)**				
Head/Neck	32 (65.3)	18 (54.5)	14 (87.5)	0.240 (ϕ)
Trunk	2 (4.1)	2 (6.1)	0 (0.0)	
Upper limbs	6 (12.2)	4 (12.1)	2 (12.5)	
Lower limbs	4 (8.2)	4 (12.1)	0 (0.0)	
Thickness (mm) ^±^	5.0 (3.0–7.25)	3.0 (2.0–5.0)	7.0 (6.0–11.0)	**0.006 *** (‡)
Diameter (mm) ^±^	1.5 (1.0–5.5)	1.10 (0.80–2.20)	2.70 (1.50–12.0)	**0.007 *** (‡)
**Clark level *n* (%)**				
I–III	9 (18.4)	9 (27.3)	0 (0.0)	**0.004 *** (ϕ)
IV–V	19 (38.8)	8 (24.2)	11 (68.8)	
**Perineural invasion *n* (%)**				
Absence	41 (83.7)	31 (93.3)	10 (62.5)	**0.010 *** (ϕ)
Presence	8 (16.3)	2 (6.1)	6 (37.5)	
**Differentiation *n* (%)**				
Moderate-Poor	17 (34.7)	7 (21.2)	10 (62.5)	0.166 (ϕ)
High	16 (32.7)	11 (33.3)	5 (31.3)	

^±^ Median (25–75th). (y): years, (m): months. (mm): millimeter. (%): frequency distribution. (‡) Mann–Whitney. (ϕ) χ2 test. * *p*-value < 0.05 in bold is estimated as significant.

**Table 2 ijms-25-00379-t002:** Univariate binary logistic regression of the main clinicopathological variables and the existence of progression.

Variable	OR (CI 95%)	*p*-Value
Age (y)		
≤75 y vs. >75 y	3.850 (1.078–13.751)	**0.038 ***
Thickness (mm)		
≤6 mm vs. >6 mm	9.333 (1.447–60.213)	**0.019 ***
Diameter (mm)		
<20 mm vs. ≥20 mm	2.154 (0.272–17.025)	0.467
Clark level		
II–II vs. IV–V	2.224 (1.082–4.571)	**0.030 ***
Perineural invasion		
Absent vs. Present	9.300 (1.613–53.618)	**0.013 ***

(y): years. (mm): millimeter. (%): frequency distribution. * *p*-value < 0.05 in bold is estimated as significant.

**Table 3 ijms-25-00379-t003:** Clinicopathological characteristics of the patients included in the study by gene expression of angiogenic and lymphangiogenic biomarkers.

	CSCC n (%)	*VEGFA* $\bar{x}$ (SEM)	*p*-Value	*VEGFR2* $\bar{x}$ (SEM)	*p*-Value	*VEGFC* $\bar{x}$ (SEM)	*p*-Value	*VEGFR3* $\bar{x}$ (SEM)	*p*-Value	*LYVE1* $\bar{x}$ (SEM)	*p*-Value	*PROX1* $\bar{x}$ (SEM)	*p*-Value
**Age at diagnosis**													
≤75 (y)	26 (53.0)	1.47 (±0.19)	0.674	1.92 (±0.61)	0.186	0.28 (±0.06)	0.280	1.83 (±0.55)	**0.043 ***	3.52 (±1.13)	0.056	1.47 (±0.62)	0.132
>75 (y)	23 (47.0)	1.72 (±0.41)		1.28 (±0.71)		0.65 (±1.15)		1.00 (±0.74)		0.91 (±0.46)		0.37 (±0.23)	
**Gender**													
Female	21 (42.8)	1.37 (±0.26)	0.585	0.90 (±0.34)	0.471	0.47 (±0.14)	0.902	2.01 (±0.95)	0.964	3.21 (±1.13)	0.257	1.52 (±0.72)	0.121
Male	28 (57.2)	1.75 (0.32)		2.16 (±0.76)		0.45 (±0.10)		1.01 (±0.35)		1.61 (±0.77)		0.53 (±0.29)	
**Disease-Free Survival**													
≤75 (m)	26 (53.0)	1.62 (±0.21)	0.179	1.55 (±0.60)	0.992	0.54 (±0.12)	0.206	1.33 (±0.70)	0.301	2.07 (±0.83)	0.574	0.74 (±0.32)	1.000
>75 (m)	23 (47.0)	1.55 (±0.40)		1.70 (±0.72)		0.36 (±0.10)		1.56 (±0.56)		2.55 (±1.05)		1.20 (±0.66)	
**Localization**													
Head/Neck	32 (65.3)	1.75 (±0.30)	0.405	1.77 (±0.57)	0.473	0.55 (±0.11)	0.133	1.07 (±0.43)	0.682	2.36 (±0.76)	0.794	1.33 (±0.52)	0.413
Trunk	2 (4.1)	1.31 (±1.31)		0.0 (±0.0)		0.0 (±0.0)		0.33 (±0.33)		0.0 (±0.0)		0.0 (±0.0)	
Superior limb	6 (12.2)	1.12 (±0.30)		0.39 (±0.35)		0.43 (0.24)		4.25 (±2.73)		0.85 (±0.45)		0.73 (±0.54)	
Lower limb	4 (8.2)	0.69 (±0.24)		0.70 (±0.69)		0.04 (±0.04)		0.84 (±0.65)		3.10 (±3.02)		0.0 (±0.0	
Unknown	5 (10.2)	1.93 (±0.54)		3.54 (±2.54)		0.43 (±0.23)		1.40 (±0.97)		3.93 (±3.81)		0.0 (±0.0)	
**Thickness ^±^**													
≤4 (mm)	11 (42.3)	1.81 (±0.37)	1.000	2.18 (±0.71)	0.109	0.31 (±0.13)	0.237	1.61 (±0.64)	0.069	1.56 (±1.49)	0.646	1.51 (±1.17)	0.760
>4 (mm)	15 (57.7)	1.73 (±0.36)		1.50 (1.05)		0.60 (±0.15)		0.42 (±0.20)		1.80 (±0.75)		0.30 (±0.24)	
**Diameter ^+^**													
≤1.5 (cm)	24 (53.3)	1.84 (±0.39)	0.632	2.11 (±0.71)	0.068	0.40 (±0.09)	0.963	1.47 (±0.54)	0.681	2.85 (±1.00)	0.411	1.32 (±0.64)	0.786
>1.5 (cm)	21 (46.7)	1.32 (±0.23)		0.71 (±0.33)		0.46 (±0.13)		1.53 (±0.86)		1.18 (±0.54)		0.71 (±0.36)	
**Clark index**													
I–III	9 (18.4)	2.01 (±0.51)	0.676	4.50 (±2.09)	0.099	0.33 (±0.16)	0.117	0.51 (±0.26)	0.567	2.42 (±2.11)	0.929	0.55 (±0.44)	0.374
IV–V	19 (38.8)	1.65 (±0.28)		0.67 (±0.20)		0.67 (±0.15)		0.94 (±0.45)		1.49 (±0.66)		0.23 (±0.19)	
Unknown	21 (42.8)	1.35 (±0.38)		1.25 (±0.43)		0.32 (±0.10)		2.29 (±0.95)		2.98 (±1.11)		1.79 (±0.75)	
**Perineural invasion**													
No invasion	41 (83.6)	1.54 (±0.25)	0.234	1.84 (±0.54)	0.781	0.37 (±0.07)	**0.030 ***	1.65 (±0.53)	0.604	2.38 (±0.76)	0.740	1.03 (±0.41)	1.000
Invasion	8 (16.4)	1.82 (±0.33)		0.52 (±0.25)		0.89 (±0.29)		0.38 (±0.25)		1.85 (±1.10)		0.57 (±0.44)	
**Differentiation**													
Moderate/Poor	17 (34.8)	1.55 (±0.26)	0.234	0.48 (±0.28)	0.162	0.70 (±0.16)	0.192	2.19 (±1.11)	0.875	1.88 (±0.92)	0.631	1.28 (±0.58)	0.371
High	16 (32.6)	1.78 (±0.26)		1.29 (±0.46)		0.49 (±0.14)		0.83 (±0.35)		2.08 (±1.12)		1.24 (±0.85)	
Unknown	16 (32.6)	1.44 (±0.56)		3.16 (±1.24)		0.17 (0.07)		1.26 (0.65)		2.96 (±1.40)		1.79 (±0.75)	

Median. (y): years, (m): month. (mm): millimeter. (cm): centimeter. (%): frequency distribution. DFS: disease-free survival. Gene expression analyses were performed to compare clinicopathological characteristics and quantitative gene expression of the target genes. ^±^ For this variable, 26 patients were available. ^+^ For this variable, 45 patients were available. *p*: *p*-value. $\bar{x}$: mean expression. SEM: standard error of the mean. * *p*-value < 0.05 was considered significant.

**Table 4 ijms-25-00379-t004:** Kaplan–Meier and Cox regression results considering tumor diameter and perineural invasion as correction factors for each gene.

Gene	Cox Regression ^±^		Log Rank Kaplan–Meier *p*-Value
	HR (95% CI)	*p*-Value	
VEGFA Diameter Perineural invasion	1.011 (0.703–1.454) 1.037 (0.988–1.088) 3.187 (1.038–9.388)	0.954 0.142 0.036 *	0.975
VEGFR2	0.879 (0.631–1.223)	0.443	0.158
Diameter	1.031 (0.982–1.083)	0.219	
Perineural invasion	2.951 (0.998–8.728)	0.050 *	
VEGFC	**2.675 (1.089–6.570)**	**0.032 ***	**0.027 ***
Diameter	1.044 (0.996–1.095)	0.072	
Perineural invasion	2.814 (0.948–8.350)	0.062	
VEGFR3	1.037 (0.861–1.250)	0.698	0.264
Diameter	1.037 (0.988–1.087)	0.140	
Perineural invasion	3.366 (1.099–10.307)	0.034 *	
LYVE1	0.923 (0.762–1.117)	0.410	0.184
Diameter	1.034 (0.958–1.084)	0.178	
Perineural invasion	3.095 (1.053–9.099)	0.040 *	
PROX1	0.895 (0.644–1.243)	0.508	0.981
Diameter	1.040 (0.989–1.094)	0.129	
Perineural invasion	3.017 (1.019–8.934)	0.046 *	

^±^: Cox regression adjusted for tumor diameter and perineural invasion as correction factors. CI: confidence interval. * *p*-value < 0.05 in bold was considered significant.

**Table 5 ijms-25-00379-t005:** Angiogenic and lymphangiogenic target genes and TaqMan probes used for qPCR detection of each gene under study.

Gene	Title	Accession No.	Amplicon Size (bp ^#^)	TaqMan Assay
*VEGFA*	Vascular endothelial growth factor A	NM_001171622	59	Hs00900055_m1
*VEGFR2*	Vascular endothelial growth factor receptor 2	NM_002253	72	Hs00911690_m1
*VEGFC*	Vascular endothelial growth factor C	NM_005429	66	Hs01099203_m1
*VEGFR3*	Vascular endothelial growth factor receptor 3	NM_002020	55	Hs01047683_g1
*LYVE1*	Lymphatic vessel endothelial hyaluronan receptor 1	NM_006691	68	Hs00272659_m1
*PROX1*	Prospero homeobox 1	NM_001270616	74	Hs00896294_m1

^#^ bp: base pair; TaqMan assay consisting of primers and specific probes (ThermoFisher Scientific Inc., Waltham, MA, USA).

## Data Availability

Data is contained within the article.

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
