# Peer review of "VEGFC Gene Expression Is Associated with Tumor Progression and Disease-Free Survival in Cutaneous Squamous Cell Carcinoma"

_ijms, 2023, doi:10.3390/ijms25010379_

Round 1
Reviewer 1 Report
Comments and Suggestions for Authors
This is a study of set of genes related to angiogenesis and lymphangiogenes is a small set of cutaneous SCC patient samples to define markers of disease progression and invasion. They have found a significant association between perineural invasion with disease progression, however, this is a modest effect and because of the small sample size, perineural invasion subset is only 8 patients. Statistical analysis of such a small patient set is unreliable, and the initial findings have not been validated in independent sample sets. They also find a weak statistical association between VEGFC expression and perineural invasion. Gene expression analysis shows a large overlap between non-invasive and invasive samples, although the protein levels determined by immunostaining shown in Figure 2 does suggest a difference. It would be more informative to show the individual patient data points rather than the bar graph. Where does the survival data in Figure 3 come from? Is this from the small patient dataset used by this group? If so, what is the criteria for high VEGFC expression? In Figure 1C there is a lot of overlap in the VEGFC expression between progressors and non-progressors, but the survival data shows a clear difference. I find it difficult to understand how this can be the case if the patients were selected solely on gene expression (data in Figure 1C), but it may be plausible if the protein levels from Figure 2 were used to define the patient populations. The data presented is at best suggestive of a role of VEGFC in CSCC progression, but this must be validated in at least one independent data set.
Comments on the Quality of English LanguageIts fine
Reviewer 2 Report
Comments and Suggestions for Authors
The current study evaluated the prognostic values of several angiogenesis and Lymphangiogenesis genes in patients with cutaneous squamous cell carcinoma (CSCC). The findings confirmed the prognostic role of several histological parameters. In addition, it found that VEGFC expression is significantly higher in patients with progressive disease and that it is significantly associated with patient disease free survival.
The study is interesting and important since the scarce of studies investigating angiogenesis and lymphangiogenesis genes in CSCC. The study is well designed, and data are nicely presented. However, I have few comments below.
Major comments:
- Please discuss why only diameter and perineural invasion were included in the cox regression model, although also age, Clarke level and thickness showed association with progression? Please discuss or include the other parameters in the model. It would be good to see whether the prognostic value of VEGFC is independent of other clinicopathological features.
- It would be interesting to test the prognostic values of genes under study in publicly available gene expression databases (hopefully with large number of patients) and discuss findings in the light of the current data.
Minor comments:
- Please revise the abbreviation throughout the manuscript and make sure they are fully defined at first mention.
- In table 4: Please revise the way numbers are presented. The decimal should be presented as dot rather than a comma (i.e: “1.3” instead of “1,3”).
- The way data of a given parameter are presented in tables should follow the style of that parameter in the table header. For example: if table header states “Mean;SEM”, data in the body of the table should be presented as “45;6” not 45(6), where “45” is the mean and “6” is SEM. Otherwise, table header should be changed to “Mean(SEM)”.
- In table 4, for CSCC column, data in parenthesis should be defined in the row names.
